# Validation of morphological ear classification devised by principal component analysis using three-dimensional images for human identification

Hitoshi Biwasaka[1,2], Akihito Usui[3], Masataka Takamiya[2], Nikolaos Angelakopoulos[4], Roberto Cameriere[5], Akiko Kumagai[6] *

1 Forensic Science Laboratory, Iwate Prefectural Police Headquarters, Iwate, Japan, 2 Division of Legal Medicine, Department of Forensic Science, Iwate Medical University, Iwate, Japan, 3 Department of Diagnostic Image Analysis, Tohoku University Graduate School of Medicine, Miyagi, Japan, 4 Department of Orthodontics and Dentofacial Orthopedics, University of Bern, Bern, Switzerland, 5 Department of Medicine and Health Science "Vincenzo Tiberio", University of Molise, Age Estimation Project, Campobasso, Italy, 6 Division of Forensic Odontology and Disaster Oral Medicine, Department of Forensic Science, Iwate Medical University, Iwate, Japan

* kumagaia@iwate-med.ac.jp

**Data Availability Statement:** All relevant data are within the manuscript files.

## Abstract

This study attempts to classify ear morphology for human identification in forensic investigations by distinguishing between the upper auricle and lobule areas. A three-dimensional homologous model of the ear was created using 414 ear images of males aged 17–93 years reconstructed from computed tomography scans of forensic autopsy cases. Morphological changes were visualized using principal component analysis and areas of significant individual differences within the entire ear were identified. The classification criterion images for the upper auricle (ten images) and lobule (12 images) were developed by combining multiple principal component values: components 1–5 for the upper auricle and 1–6 for the lobule. Three-dimensional ear images of the upper auricle and lobule areas from 414 subjects were categorized using a measurement method based on the minimum distance between 5,507 corresponding points. The results indicate the applicability of the criterion images for the morphological classification of ears in this study.

## Introduction

Facial feature-based human identification in forensic science involves comparing morphological classifications or indices such as facial features, eyebrows, eyes, nose, lips, and ears. In particular, the ears exhibit wide individual differences [1–3]. Even when the face is covered by a mask, the ears can serve as a useful identification feature, as long as they are not hidden by hair. Although morphological classifications exist for specific parts of the ear such as the helix, antihelix, tragus, scaphoid fossa, and lobe, a comprehensive morphological classification of the entire ear shape has not yet been established. Developing such a classification would allow

**Funding:** The author(s) received no specific funding for this work.

**Competing interests:** The authors declare that they have no known competing financial interests or personal relationships that could appeared to influence the work reported in this paper. The authors have declared that no competing interests exist.

morphological similarities to be quantified, enabling objective evaluation for human identification.

Cameriere et al. divided the photographs of Italian ears into four areas (helix, antihelix, concha, and lobe) using simple and reproducible points and lines based on anatomical features. They analyzed the correlation between the ratios of these areas and identified individual differences, suggesting that these findings may be common across different races [4, 5]. Therefore, the establishment of a morphological ear classification system would be beneficial for forensic investigations on an international scale. Additionally, in our previous study, we used principal component analysis (PCA) [6–8], a method employed for age and sex discrimination of pelvic bones [9, 10], and pattern recognition of facial images [11] to develop ear classification criterion images (CCIs) from three-dimensional (3D) images of Japanese male ears reconstructed from computed tomography (CT) scans. As a result, their effectiveness in human identification was suggested [12]. However, owing to the large individual differences in lobule shape, the characteristics of the lobule significantly influenced the CCIs. Therefore, we re-evaluated the number of subjects and principal components (PCs) required to devise ear CCIs and created new CCIs by separately analyzing the upper auricle and lobule areas.

## Materials and methods

### Reconstruction of 3D ear images

CT images of 414 male cadavers (aged 17–93 years; mean, 53.3; standard deviation, 20.5) obtained within seven days postmortem and exhibiting non-deformed ears at the time of imaging were collected from forensic autopsies conducted at the Iwate Medical University and Tohoku University during 2009–2013 (Table 1). Each analysis in this study was conducted using varying numbers of participants to determine the optimal number of models required to develop the final CCIs (Table 2). The left and right ear regions were extracted from the head skin image reconstructed using the skin-side CT value (approximately –500 HU) via surface rendering with OsiriX MD software (version 12.5.1; Pixmeo, Geneva, Switzerland) from digital imaging and communications in medicine (DICOM) data and converted into 3D files (STL data) [13]. The right ears were mirrored using HBM-Rugle software (Medic Engineering Corp., Kyoto, Japan), and the orientation of all ear images was standardized.

### Homologous modeling for 3D ear models

A 3D ear image of a man in his 40s was used as the template model, comprising 5,507 corresponding points, 16,521 polygons, and 18 landmarks. The template model was aligned with the same landmarks as other 3D ear images, deformed, and adjusted to establish connectivity information. The 3D ear model was created using homologous modeling with Markerless

**Table 1. Distribution of age, left and right ear of all CT data.**

| Age groups (years) | Left | Right | Total |
|---|---|---|---|
| 17–29 | 34 | 29 | 63 |
| 30–39 | 35 | 35 | 70 |
| 40–49 | 30 | 24 | 54 |
| 50–59 | 28 | 29 | 57 |
| 60–69 | 35 | 29 | 64 |
| 70–79 | 27 | 25 | 52 |
| 80–93 | 29 | 25 | 54 |
| Total | 218 | 196 | 414 |

**Table 2. Distribution of age, left and right ear of CT data for difference of each set of models.**

| Age groups (years) | Number of models in set | | | | | | |
|---|---|---|---|---|---|---|---|
| | 123 | 242 | | 300 | | 363 | |
| | Left* | Left | Right | Left | Right | Left | Right |
| 17–29 | 20 | 22 | 18 | 23 | 19 | 25 | 22 |
| 30–39 | 18 | 19 | 17 | 25 | 24 | 29 | 28 |
| 40–49 | 18 | 19 | 16 | 22 | 19 | 26 | 21 |
| 50–59 | 16 | 16 | 16 | 22 | 22 | 27 | 28 |
| 60–69 | 16 | 19 | 13 | 27 | 21 | 35 | 30 |
| 70–79 | 14 | 14 | 11 | 16 | 15 | 27 | 16 |
| 80–93 | 21 | 23 | 19 | 25 | 20 | 22 | 27 |
| Total | 123 | 132 | 110 | 160 | 140 | 191 | 172 |

*: In the set with 123 models, only the left ear of each subject was extracted to avoid the risk of testing the same individuals whose shapes were similar on both sides given the small number of subjects.

Homologous Body Modeling software (mHBM, Medic Engineering Corp., Kyoto, Japan) [14–16] (Fig 1). The upper auricle model, which excluded the lobule area connecting the cauda helicis, incisura intertragica, and otobasion inferius, and the lobule-only model were separately generated using HBM-Rugle. Subsequently, the upper auricle and lobule-only models were analyzed individually (Fig 2).

## Principal component analysis (PCA)

PCA of the ear models was conducted using DHRC-HBS-PCA™ software developed by the National Institute of Advanced Industrial Science and Technology, Digital Human Research Center, Tsukuba, Japan. PCA is used to reduce a large amount of variable data to a smaller set of variables and principal components (PCs), while minimizing information loss [6]. The software was specifically designed to identify shape components of the human body using PCA and to generate a 3D image based on the obtained PC values [14–16]. To exclude the effect of ear size variation, an average image of all ear models was created, and PCA was performed

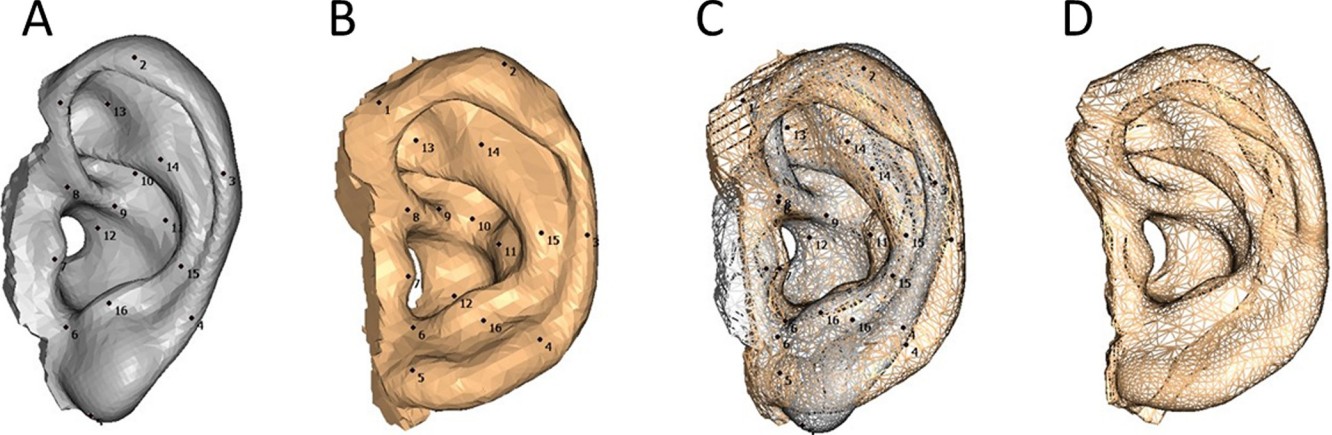

**Fig 1. Creation of a homologous ear model.** (A) Template model using the 3D ear image of a 40s-man, 5,507 corresponding points, 16,521 polygons, and 18 landmark points. (B) Another 3D ear image with the same 18 landmark points. (C) A superimposition image of A and B using 18 landmark points. The vertices of the template model were adjusted to those of the 3D image. (D) Completed homologous ear model.

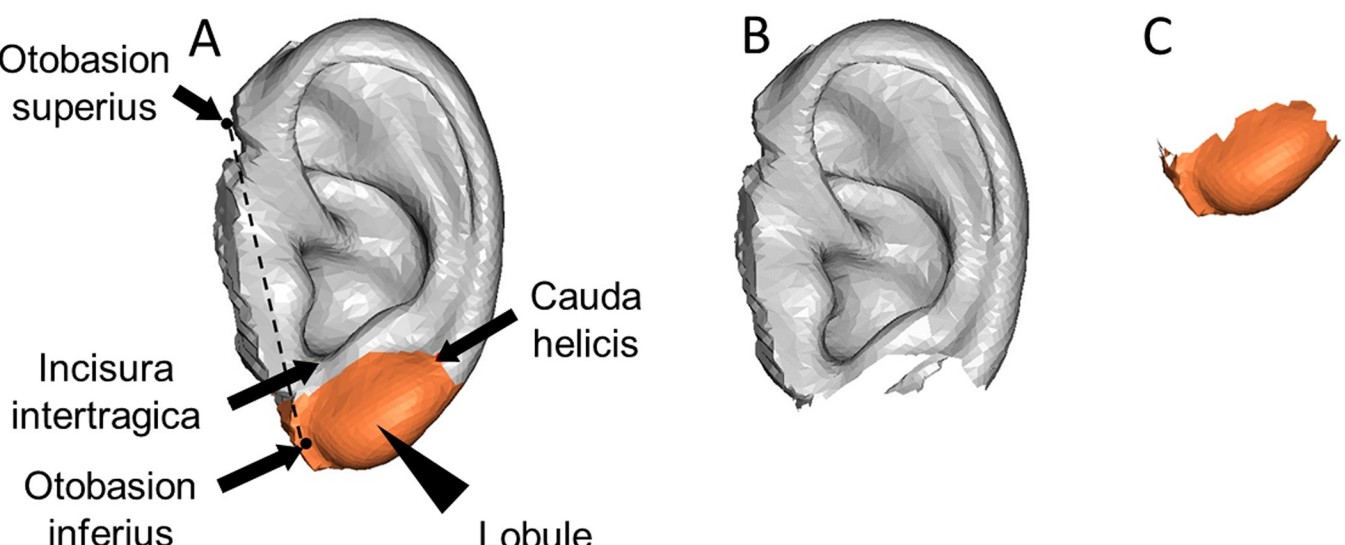

**Fig 2. Separation of the upper auricle and lobule from the homologous model.** (A) The average image of a homologous model. (B) The upper auricle homologous model after removing the lobule area surrounded by the cauda helicis, the lower part of incisura intertragica, and otobasion inferius. (C) A homologous model for the lobule only.

after standardizing the size and posture of each model to match those of the average image using HBM-Rugle software.

## Determination of the number of PCs and number of models in the set to devise CCIs

To devise the CCIs, it is important to consider the differences in the shapes of each subject. As the number of PCs and models used in the analysis significantly affected the shape difference, the following method was used to confirm the shape change area and determine the appropriate number of PCs and models.

1. Scree plot method
   The PC number was identified immediately before the curve changed from steep to smooth in the eigenvalue (EV) graph, using PCA for each set of models [17].

2. Measurement of distance between vertices (DBV)
   The DBV in ear models was created by applying both minus three times and plus three times the standard deviation (SD) value of each PC value detected by PCA was measured using the HBM-Rugle software to identify the PC number with the largest distance [18, 19].

3. Observation of shape change area
   The ear models created with three times the SD value by the PCA of each set of models were superimposed on the average ear model to search for the PC number in which the changed area was the same using the HBM-Rugle software.

## Devising of upper auricle and lobule CCIs

A composite image (CI) was generated from a scatter biplot that combined two PCs, ensuring that all PCs were inherent as shape components in the CCIs. Following PCA of the composite images to reduce dimensionality, the CCIs for the upper auricle and lobule were developed using selected valuable PCs identified using the scree plot method.

### Verification of the accuracies of CCIs in distinguishing ear morphology

The DBV between each CCI and the 3D ear images of 414 subjects was measured using the HBM-Rugle software to validate the applicability of the devised CCIs in distinguishing ear morphology. The image with the CCI showing the minimum DBV was selected as the classification image. In cases in which multiple CCIs had DBV values below the threshold, the CCI most visually similar to the image was selected. Images with a DBV greater than the threshold value were excluded from the classification process.

This study was approved by the Ethical Review Committee Iwate Medical University (Approval No. 01354) and conducted in accordance with the ethical standards outlined in the Declaration of Helsinki. Because the informed consent from the subjects cannot be obtained, the ethical committee waived the requirement of the collecting informed consent. The opt-out method ensured that bereaved family members had the opportunity to refuse participation. The all collected data were anonymized before we accessed them, except sex, date of birth, and the date of taking CT images. The data were accessed for research purposes between 1st and 30th September 2023. An extreme care was taken to protect leakage of personal information during whole of this study.

## Results

### PCA by the numbers of models in the set

Tables 3 and 4 show the correlations between PC, EV, contribution ratio (CR), and age in the PCA of the 414 upper auricle and lobule models. Twenty-four PCs were detected in the upper auricle model and the difference in EV was greater than 500 until PC6. The cumulative contribution ratio (CCR) was 74.7%. The software only displayed PCs with a CR of 1% or higher, which means that there were many PCs with a CR of 1% or lower, suggesting the complexity of the auricular structure. The number of PCs detected were 25, 25, 24, and 24 PCs in the 123, 242, 300, and 363 models, respectively. Fig 3 shows the shape change in the ear model created using the standard deviation (SD) for PC1, which had the largest EV in the PCA. Shape changes in the antihelix and helix were particularly significant.

Seventeen PCs were detected in the lobule model, and the difference in EV was > 100 until PC6. CCR was 91.0%. The number of PCs detected were 15, 16, 16, and 17 PCs in the 123, 242, 300, and 363 models, respectively.

There was no significant correlation between PC and age.

### Number of models and PCs for creating CCIs

In the scree plot curves of the 123, 242, and 300 models of the upper auricle, EV showed a steep slope down to PC5, and the 363 and 414 models showed a steep slope down to PC6 (Fig 4). The DBV showed that the shape changes in the 123, 242, and 300 models were gradual and no abrupt changes were observed, whereas the 363 and 414 models showed large shape changes up to PC6, and the changes in the 363 and 414 models were similar (Fig 5). Based on these results, we decided to use PC5 and higher to devise the upper auricle CCIs.

The scree plot of the lobule model showed a steep EV slope down to PC6 for all sets of the models (Fig 6). The DBV also exhibited a large change in shape up to PC6 (Fig 7). Thus, it was considered reasonable to adopt up to PC6 for the creation of the lobule CCIs.

The areas that changed from PC1 to PC5 in > 300 upper auricle models were almost identical (Fig 8). The Lobule model exhibited the same changes up to PC6 in more than 300 models (Fig 9). Based on these results, the adoption of PC1–5 for the upper auricle and PC1–6 for the lobule to devise the CCIs was determined. As the shape-change areas of both the upper auricles

**Table 3. Result of PCA of 414 upper auricle models and the correlation between distributions of PC score and age.**

| PC | EV | CR (%) | CCR (%) | R |
|---|---|---|---|---|
| 1 | 1437.6 | 10.1 | 10.1 | −0.01 |
| 2 | 1212.3 | 8.5 | 18.5 | −0.21 |
| 3 | 1000.8 | 7.0 | 25.5 | 0.27 |
| 4 | 900.3 | 6.3 | 31.8 | −0.27 |
| 5 | 709.0 | 5.0 | 36.8 | 0.16 |
| 6 | 572.0 | 4.0 | 40.8 | 0.11 |
| 7 | 483.4 | 3.4 | 44.2 | −0.09 |
| 8 | 449.5 | 3.1 | 47.3 | 0.18 |
| 9 | 390.3 | 2.7 | 50.1 | −0.13 |
| 10 | 358.2 | 2.5 | 52.6 | 0.04 |
| 11 | 347.4 | 2.4 | 55.0 | 0.00 |
| 12 | 335.7 | 2.3 | 57.4 | 0.16 |
| 13 | 277.1 | 1.9 | 59.3 | 0.07 |
| 14 | 269.6 | 1.9 | 61.2 | −0.08 |
| 15 | 258.1 | 1.8 | 63.0 | 0.04 |
| 16 | 241.0 | 1.7 | 64.7 | 0.15 |
| 17 | 221.2 | 1.5 | 66.2 | −0.10 |
| 18 | 205.9 | 1.4 | 67.7 | −0.03 |
| 19 | 198.8 | 1.4 | 69.1 | 0.09 |
| 20 | 184.0 | 1.3 | 70.3 | 0.06 |
| 21 | 170.4 | 1.2 | 71.5 | −0.01 |
| 22 | 158.1 | 1.1 | 72.6 | −0.09 |
| 23 | 153.1 | 1.1 | 73.7 | −0.06 |
| 24 | 144.2 | 1.0 | 74.7 | −0.05 |
| **Total** | 14289 | 74.7 | | |

PCA, principal component analysis; PC, principal component; EV, eigen value; CR, contribution ratio; CCR, cumulative contribution ratio; R, correlation coefficient.

and lobules were almost identical for the sets with 300 or more models, it was decided that those larger than 300 were necessary to devise the CCI.

## Creation of CIs for determination of ear CCIs

Fifty Cls of the upper auricle models were synthesized from a combination of the two models using all CIs of PC1–5. Ten CIs synthesized with two PC values (-100 and +100) from a single PC, and forty CIs synthesized with two PC values (-100 and +100) from two different PCs (Figs 10–12).

PCA was performed to reduce the dimensionality of the 50 CIs. A scree plot of the detected PCs and EVs is shown in Fig 13. Five PCs were detected in the PCA of CIs created from 300 models, and their EVs changed gently. Six PCs were detected in the PCA of CIs created from 363 and 414 models, and their EVs decreased sharply at PC6. However, PC6 had a small EV. Therefore, PC values up to PC5 were used to devise the final CCIs.

As almost all PC values (PC1–5) of PCA by 50 CIs showed within ±100, a total of 10 CCIs were devised using single PC values (±100) of PC1–5 of three sets of models of the upper auricle. CIs from the 300 and 363 models showed a shape discrepancy; however, CIs from the 363 and 414 models were very similar, with a small shape difference of 0.6–1.6 mm (mean 1.1 mm)

**Table 4. Result of PCA of 414 lobule models and the correlation between distributions of PC score and age.**

| PC | EV | CR (%) | CCR (%) | R |
|---|---|---|---|---|
| 1 | 399 | 17.9 | 17.9 | 0.19 |
| 2 | 314 | 14.1 | 32.0 | −0.33 |
| 3 | 254 | 11.4 | 43.4 | −0.29 |
| 4 | 220 | 9.9 | 53.3 | 0.06 |
| 5 | 147 | 6.6 | 59.9 | 0.07 |
| 6 | 110 | 4.9 | 64.8 | 0.04 |
| 7 | 99 | 4.5 | 69.3 | 0.00 |
| 8 | 76 | 3.4 | 72.7 | −0.03 |
| 9 | 72 | 3.2 | 75.9 | 0.01 |
| 10 | 58 | 2.6 | 78.5 | −0.03 |
| 11 | 50 | 2.2 | 80.8 | 0.05 |
| 12 | 48 | 2.2 | 82.9 | −0.01 |
| 13 | 38 | 1.7 | 84.7 | 0.03 |
| 14 | 34 | 1.5 | 86.2 | 0.03 |
| 15 | 32 | 1.4 | 87.6 | 0.06 |
| 16 | 29 | 1.3 | 88.9 | −0.14 |
| 17 | 25 | 1.1 | 90.0 | 0.08 |
| 18 | 22 | 1.0 | 91.0 | −0.02 |
| **Total** | **2226** | **91.0** | | |

PCA, principal component analysis; PC, principal component; EV, eigen value; CR, contribution ratio; CCR, cumulative contribution ratio; R, correlation coefficient.

between the corresponding images (Fig 14). This result indicates that more than 363 models are necessary to devise the final upper auricle CCIs, and that the 414 subjects in this study are appropriate.

For all combinations of PC1–6 of the lobule models, 72 CIs were created for single and multiple principal component shapes. Fig 15 shows scatter biplots up to PC1–2 with 414 lobule models. As with the upper auricle models, twelve CIs synthesized with two PC values (-60 and +60) from a single PC, and fifty CIs synthesized with two PC values (-60 and +60) from two different PCs. Subsequently, a PCA of the 72 CIs was performed for dimensionality reduction. Twelve CCIs were devised with only a single PC shape based on single PC values (±30) of PC1–6 of the PCA of 72 CIs in the same way as the upper auricle.

## Determination of ear CCIs of the upper auricles and lobules

The final CCIs are shown in Fig 16. CCIs II, IV, and VII of the upper auricles show a strong overhang of the auricle toward the head. The CCIs of the lobules have almost the same shape; however, they were divided into those with strong lateral protrusions and those hanging downward. There are also isolated, adherent, and intermediate types of attachments to the skin.

## Accuracy of CCIs

The results of classifying the upper auricle and lobule images of 414 of the subjects as CCIs in the present study are illustrated in Table 5. Of the images, 91.3% were classified as one of the upper auricle CCIs within the threshold, and 8.7% were outside the threshold (out of classification). There were 63.5% of images with multiple CCIs detected within the threshold; however,

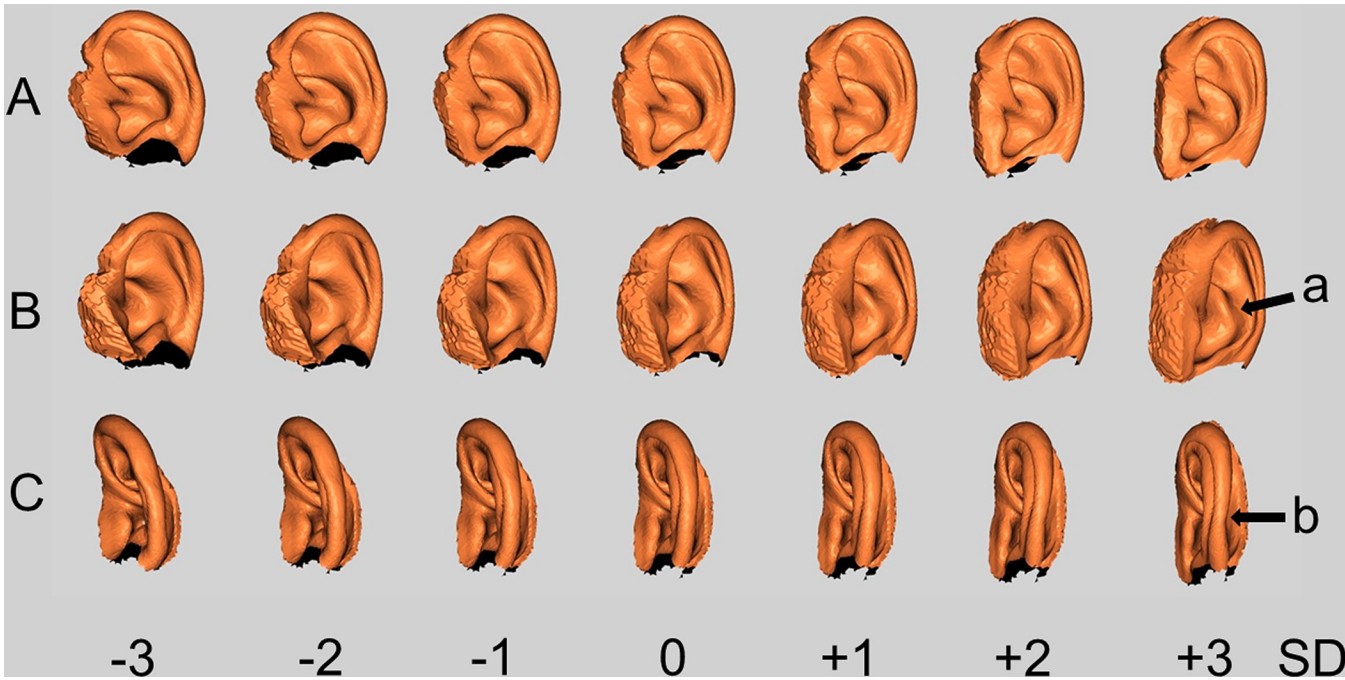

**Fig 3. Visualization of morphological changes from minus three times to plus three times of the SD value of PC1 detected in PCA of 414 upper auricle models.** SD, standard deviation; PC, principal component; PCA, principal component analysis. (A) Side images. (B) Rotated 40 degrees backward for A. (C) Rotated 40 degrees forward for A. Morphological changes of protrusion of antihelix (arrow a) and curvature of helix (arrow b) are characteristic.

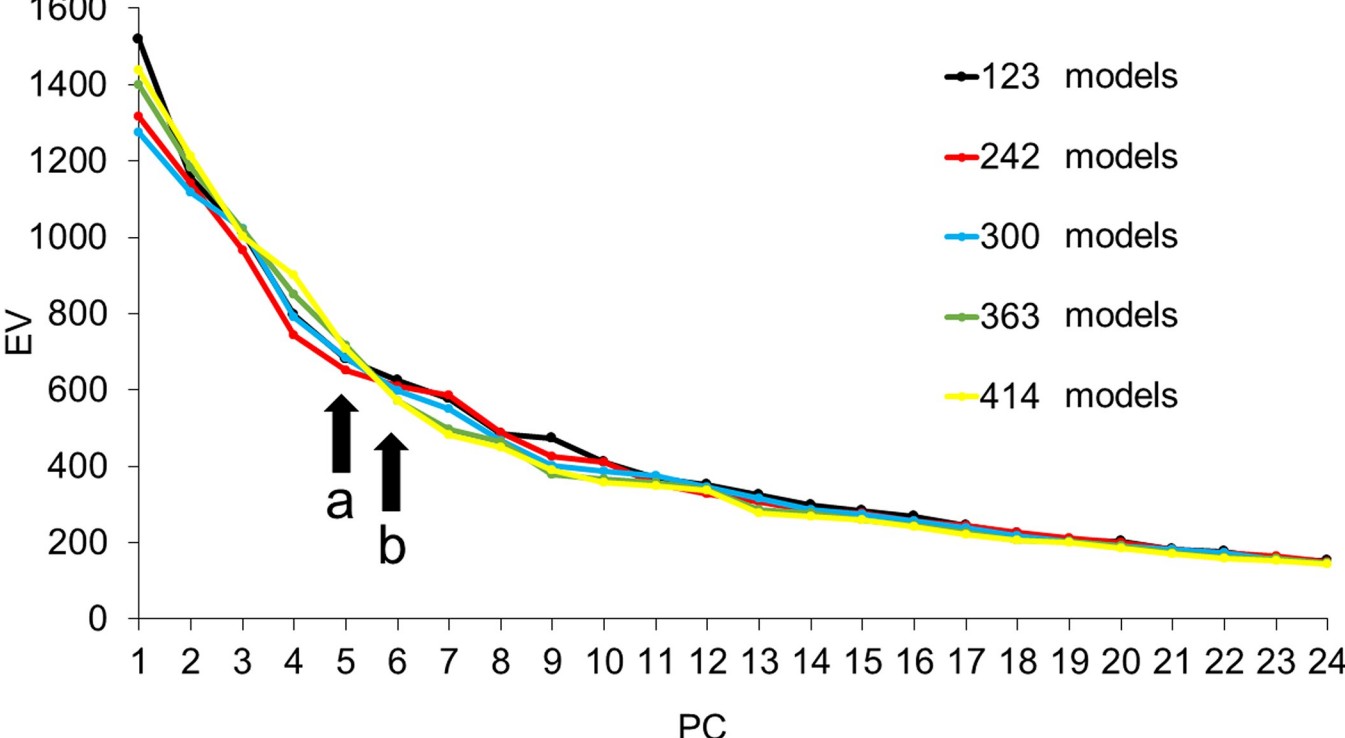

**Fig 4. Scree plot curve for five sets of models for the upper auricle.** PC, principal component; EV, eigen values. The EVs rapidly decrease until PC5 for the sets with 123, 242, and 300 models (arrow a) and PC6 for the sets with 363 and 414 models (arrow b).

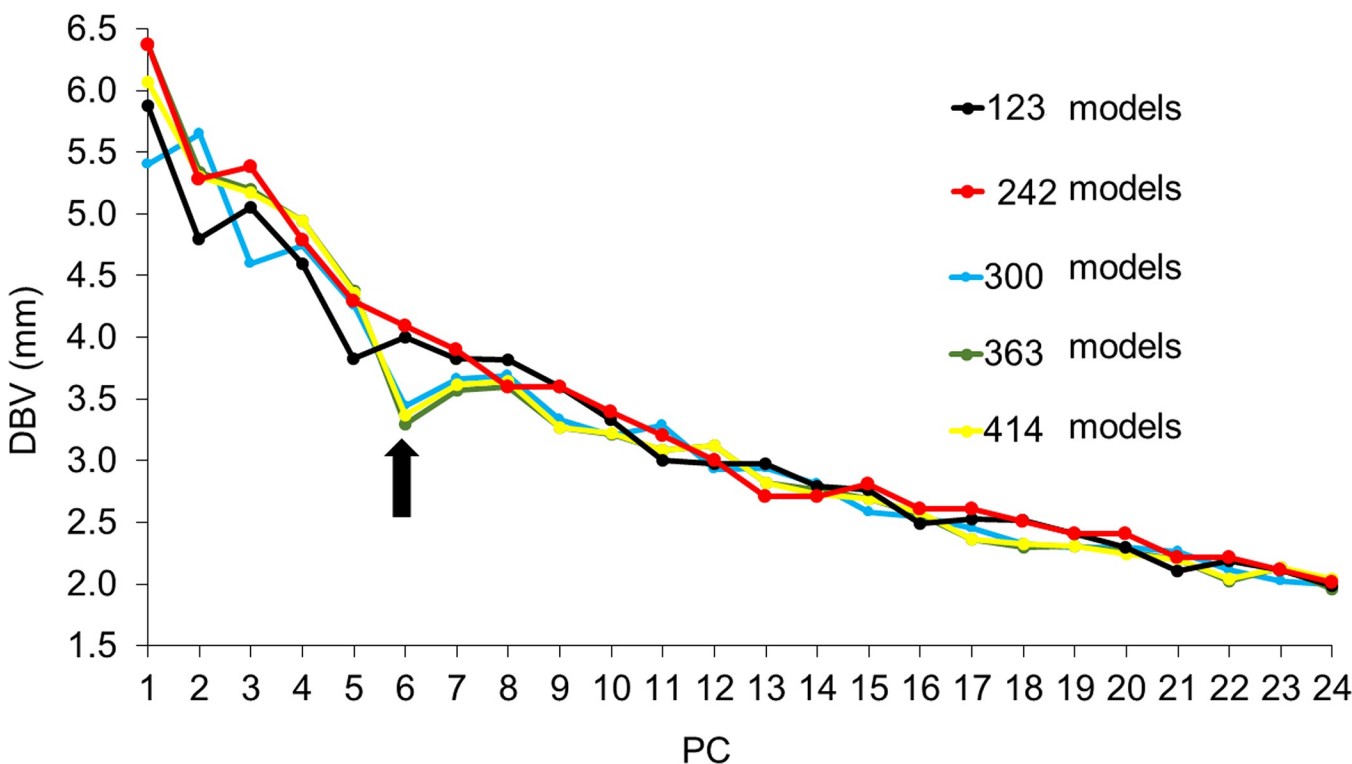

**Fig 5. Slope of the DBV of two images created by minus three times and plus three times of the SD value by each PC for five sets of models for upper auricle models.** PC, principal component; DBV, distance between vertices; SD, standard deviation. The DBV of the sets with 300, 363, and 414 models dropped significantly until PC6 (arrow), and the gradient of the plots of 363 and 414 models was well approximated.

most of those determined by macroscopy were ranked first or second in the DBV. The frequency of CCIs within the thresholds for each CCI ranged from 7% to 12%, with no significant bias (Fig 17). In contrast, 96.1% of the images were determined to be one of the lobule CCIs within the threshold, and 3.9% were outside the threshold (out of classification). There were 74.9% of images with multiple CCIs detected within the threshold; however, as with the upper auricle, most of those determined by macroscopy were ranked first or second in the DBV. The frequency of CCIs within the threshold for each CCI ranged from 3% to 15%, showing a difference in the number of determinations compared to the upper auricle (Fig 18). In addition, many images of both the upper auricle and lobule were classified as separate left and right CCIs, suggesting that human ear morphology not only differs between individuals, but also has significant left–right differences.

## Discussion

Ears, like faces, vary in morphology between individuals and are useful in criminal investigations and the forensic identification of unidentified individuals. For the classification of facial morphology, the reports by Poch's 10-type classification and Garson's face index have long been used [20]. However, due to the complexity of ear morphology, no attempt has been made to classify the entire ear. Therefore, we created complex ear CCIs by adopting a new statistical method using 3D images.

The analysis employed in the present study differed from those typically utilizing only two variables [6–8, 11], in which the PCA analyzed three variables of 3D coordinates. Therefore, the obtained PC consisted of the coordinate values of 3D shapes [14–16], and the associated

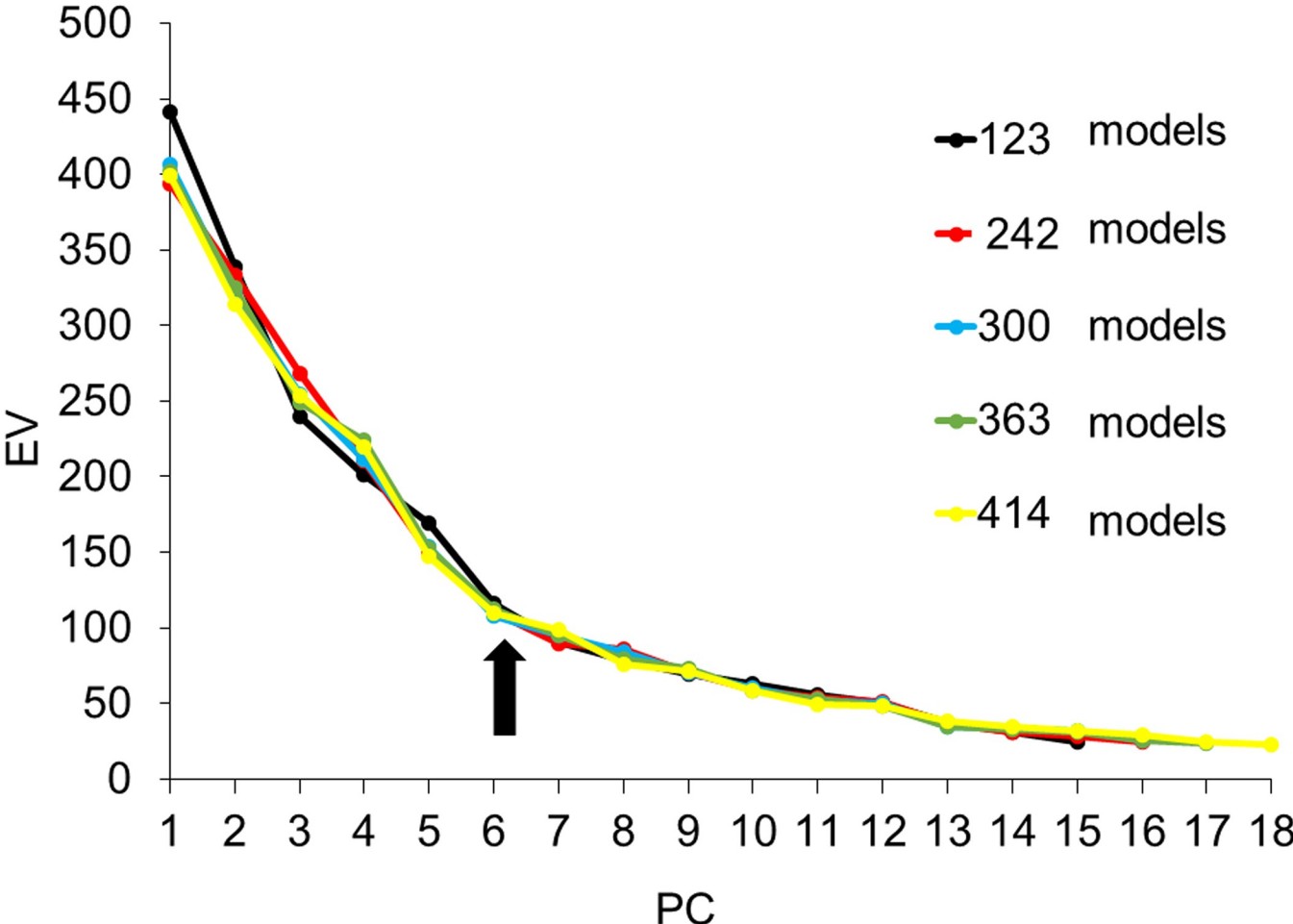

**Fig 6. Scree plot curve of five different sets of models for the lobule.** PC, principal component, EV, eigen values. The EVs decreased sharply for all sets of models until PC6 (arrow).

software had features that could help easily visualize the shape change indicated by each PC. The authors applied this method to the sex and age estimation of the pelvis and found that the number of shape points corresponding to the template model significantly affected the reproducibility of the CT image [9, 10]. Owing to the complex structure of ears, a certain number of shape points is necessary for the template model. In this study, a CT image of a man in his 40s, which was converted to 3D, was used, and the number of shape points on the ear template was 5,507 points. The number of shape points on the large pelvis reported earlier was 8,471, and the number of shape points on the ear template in this study was sufficient, considering both the pelvis and ear size [10]. In general, increasing the number of shape points on the template may increase the reproducibility of fine irregularities and the number of detected PCs. However, as shown in Figs 8 and 9, the valid PC with a larger EV indicated almost the same shape.

The authors previously generated two sets of 3D ear models, one including lobules and one without lobules, from 65 Japanese males, and developed CCIs using PCs detected by PCA. Consequently, the difference in shape between the CCIs of the two groups was minimal [12]. The lobule, which contains adipose tissue, is more susceptible to posture and external forces than other ear areas; therefore, this study was conducted separately for lobule analysis.

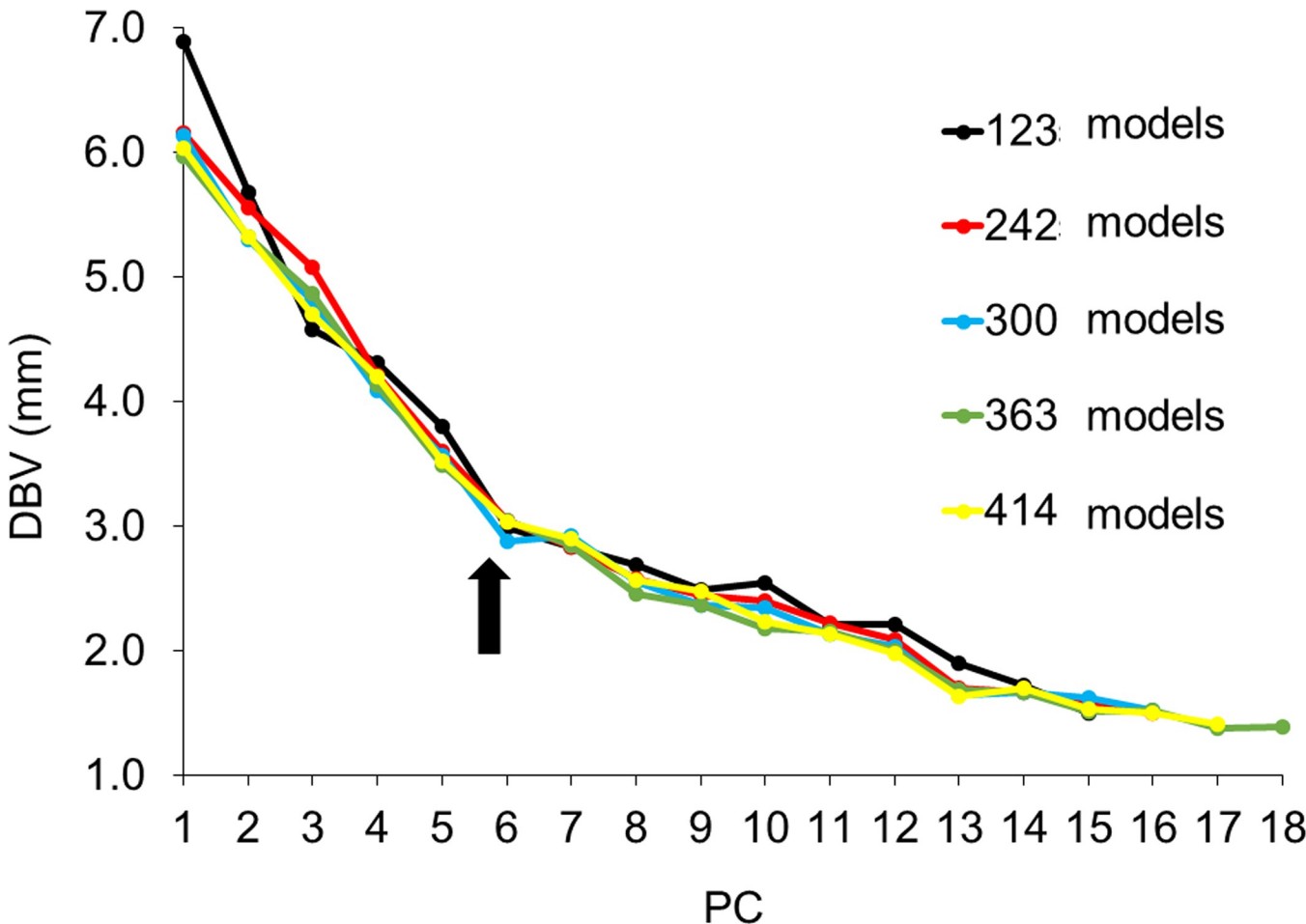

**Fig 7. Slope of the DBV of two images created by minus three times and plus three times the SD value of each PC of five different sets of models for the lobule.** PC, principal component; DBV, distance between vertices; SD, standard deviation. The DBV of all sets of models dropped significantly until PC6 (arrow).

To devise the CCIs, the authors examined the PC responsible for shape differences and the numbers of models in a set that influenced the same region of shape change. Using the scree plot method [17], the PC with a significant reduction in EV was identified (Figs 4 and 6). With the two newly projected methods using the HBM-Rugle software, the number of PCs with sharply decreasing shape changes can be identified from the DBV (Figs 5 and 7), and the necessary number of models in the set can be determined from the visualized shape change area (Figs 8 and 9). The scree plot method visually selects a PC based on the decrease in EV; however, selection becomes challenging when the slope is gradual because it is influenced by the number of subjects used. In a previous study [12], PC selection was challenging because of the limited number of participants. In the present study, a more distinct decrease in the range was observed by increasing the number of participants, allowing for a clearer projection of the number of PCs needed to devise CCIs. Therefore, the effectiveness of this method is satisfactorily demonstrated.

Anthropometric studies on auricular morphology have been reported previously [21]. However, the lack of pioneering reports on the creation of CCIs has led to the development of a new method for ear CCIs. To devise CCIs using all PCs that were adopted, the distributions of the two PCs were graphed in a biplot, and it was found that none of the PCs showed a highly

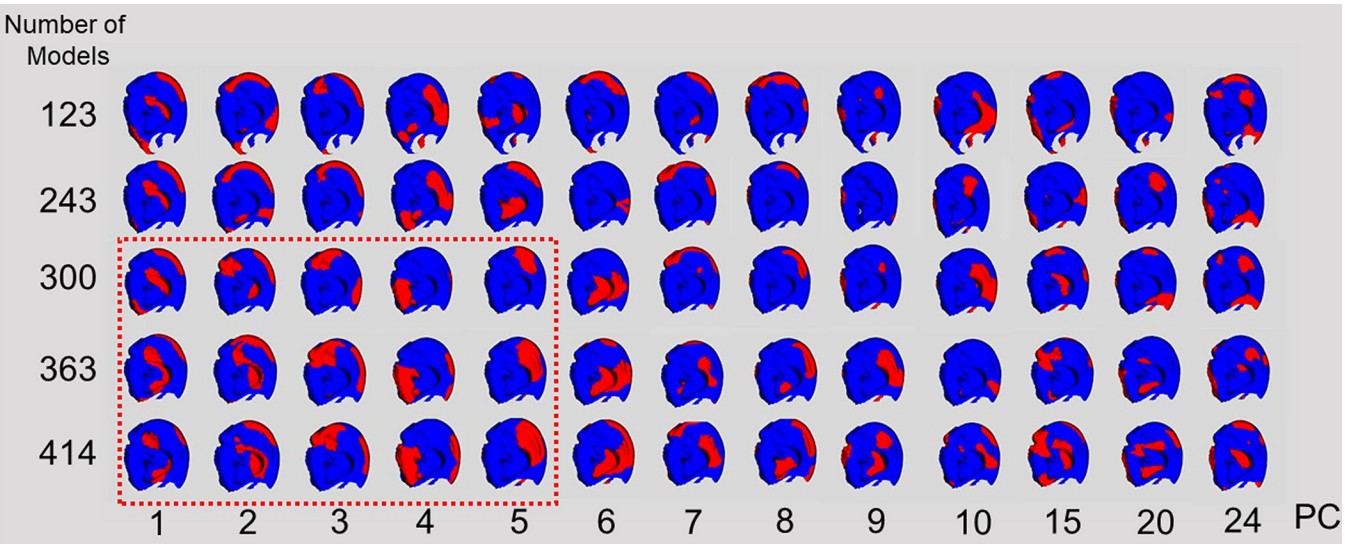

**Fig 8. Visualization of shape change upper auricle models by superimposition between the average auricle model and those created by three times the SD value for each PC.** PC, principal component; SD, standard deviation. The shape changes (red regions) in 363 and 414 models are similar until PC5 (indicated by the red dotted line).

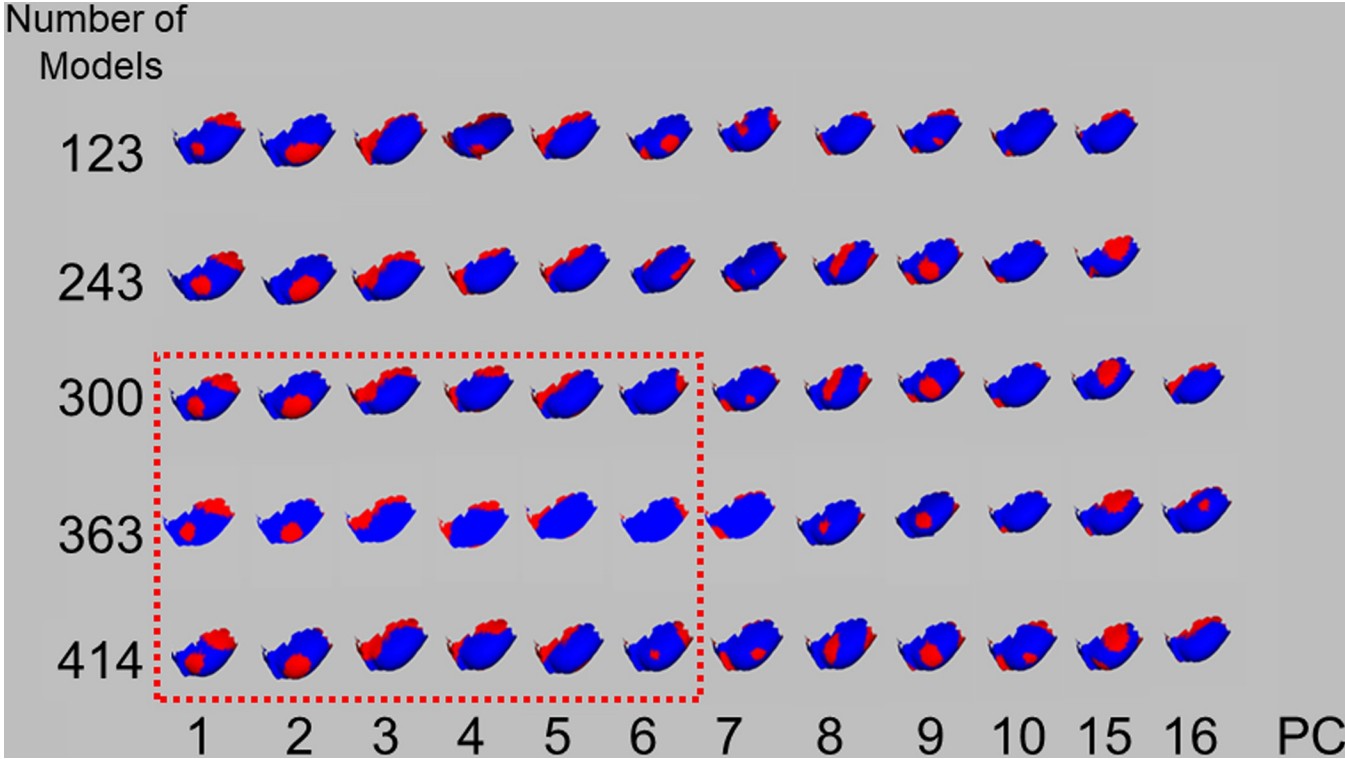

**Fig 9. Visualization of shape change lobule models by superimposition between the average lobule model and those created by three times of the SD value.** PC, principal component; SD, standard deviation. Shape changes (red regions) of 300, 363, and 414 models are similar until PC6 (indicated by the red dotted line).

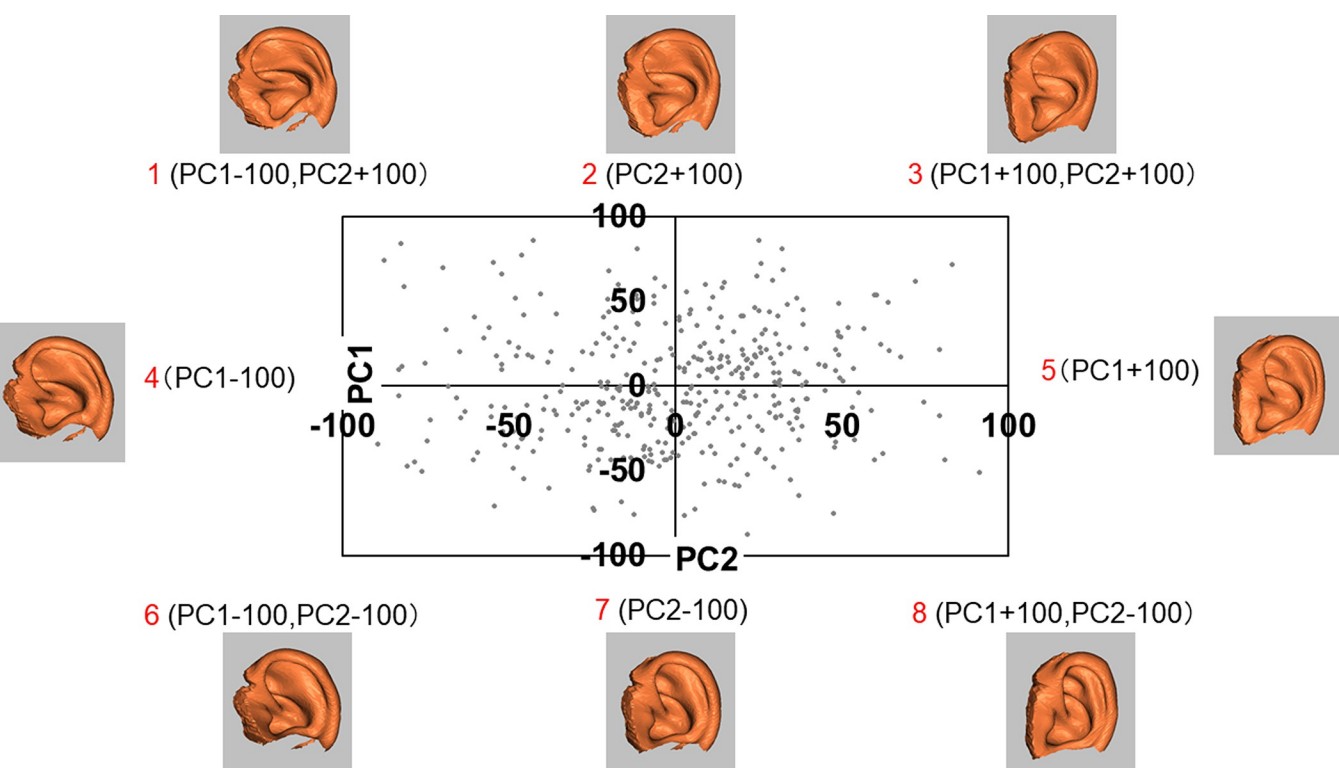

**Fig 10. Biplot of PC1 and PC2 used for synthesizing the CIs.** PC, principal component; CI, composite image. Eight upper auricle models are synthesized using single PC values (-100 or +100) of four (2, 4, 5, 7) and two different PC values of four (1, 3, 6, 8) to synthesize the CIs.

skewed scatter. Therefore, each CI was created using the values of a single PC and each of the two PCs. PCA was then repeated for dimensionality reduction, and the scree plot method was performed, in which PCs up to PC5 for the upper auricle and up to PC6 for the lobule were utilized for to create CCIs.

Although not reflected in this report, the authors previously examined 414 lobule determinations with 8, 10, and 12 CCIs and found that 12 CCIs had the lowest out-of-classification number and eight CCIs had the highest out-of-classification number. Perhaps because the lobule is generally small in size, and the mean value of the DBV between images and each CCI was used for image determination in the present study, it is possible that many subjects fell

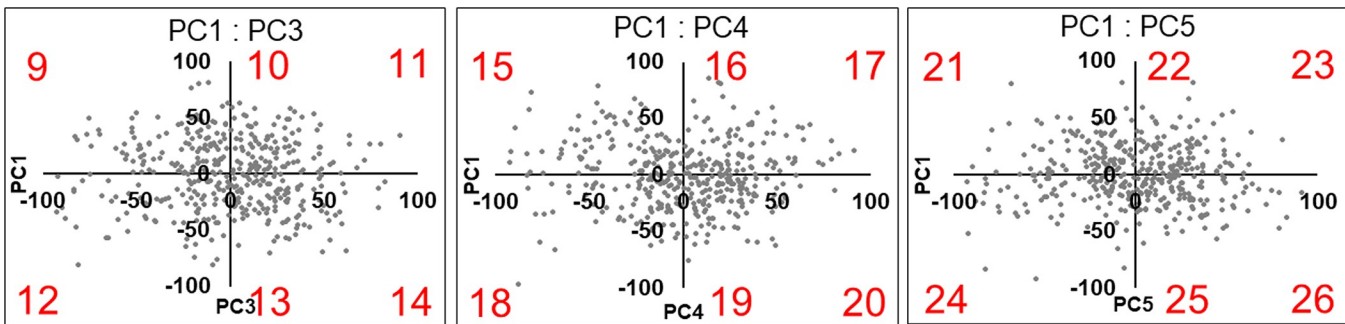

**Fig 11. Biplots of PC1 and PC3 (left), PC1 and PC4 (center), PC1 and PC5 (left) used to synthesize the CIs.** PC, principal component; CI, composite image. Eighteen upper auricle models are synthesized using single PC values of six (10, 13, 16, 19, 22, 25) and two different PC values of twelve (9, 11, 12, 14, 15, 17, 18, 20, 21, 23, 24, 26).

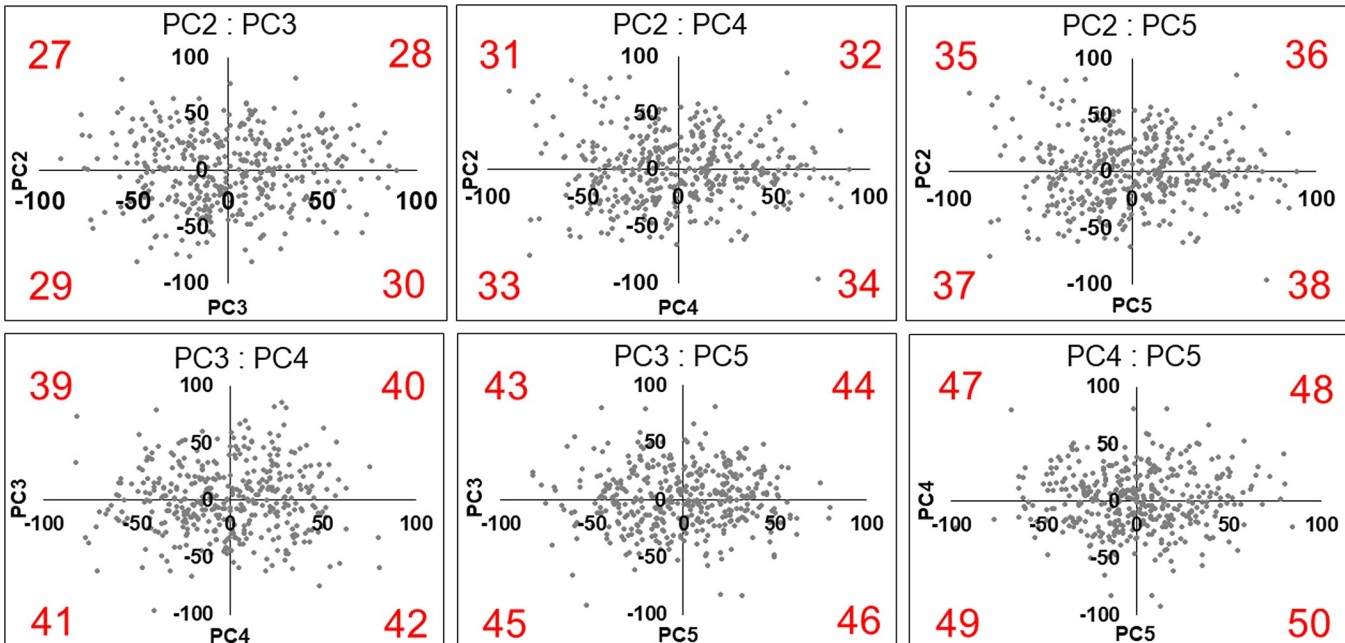

**Fig 12. Biplots of PC2 and PC3 (upper left), PC2 and PC4 (upper center), PC2 and PC5 (upper right), PC3 and PC4 (lower left), PC3 and PC5 (lower center), PC4 and PC5 (lower right) used to synthesize the CIs.** PC, principal component; CI, composite image. Twenty-four upper auricle models are synthesized using two different PC values of twenty-four (27–50).

within the threshold. To avoid out-of-classification, it is necessary to test other image determination methods such as cluster analysis using PCs. Moreover, among the CCIs shown in Fig 16, those with a strong auricle overhang may cause errors in the DBV results, particularly because of the thickness of the soft tissue on the posterior (head side) surface of the auricle. In addition, the DBV method may not accurately evaluate the similarity of the 3D ear structure to the lateral face; therefore, the establishment of CCIs for the lateral 3D ear is needed.

To utilize CCIs for live human identification, it is necessary to obtain 3D ear images of subjects. However, obtaining CT images of individuals is not always feasible; therefore, the use of ear images captured with a 3D facial imaging device has already been employed by the Japanese police [22]. If two-dimensional photographic ear images are used, the current metrological comparison methods cannot be applied. Nevertheless, based on the results of this study, a macroscopic classification of the three major morphological groups could be implemented:

1. Long upper and lower with a deep triangular fossa (Groups II, IV, V)

2. Rounded lateral view of the helix (Groups III, VI, VIII, X)

3. Wide upper antihelix with a shallow triangular fossa (Groups I, VII, IX)

However, the development of morphological ear classification in the forensic field requires ongoing research as there are still numerous potential issues to address.

The subjects of the present study were limited to males, as most of the corpses requiring post-mortem imaging diagnosis were male. To verify the effectiveness of this study on female subjects, it will be necessary to expand its scope beyond postmortem CT imaging. In this study, we reduced the variation in age and verified the number of models required for analysis. Considering the principles of PCA, we believe that CCIs that encompass the auricular features of Japanese males were generated.

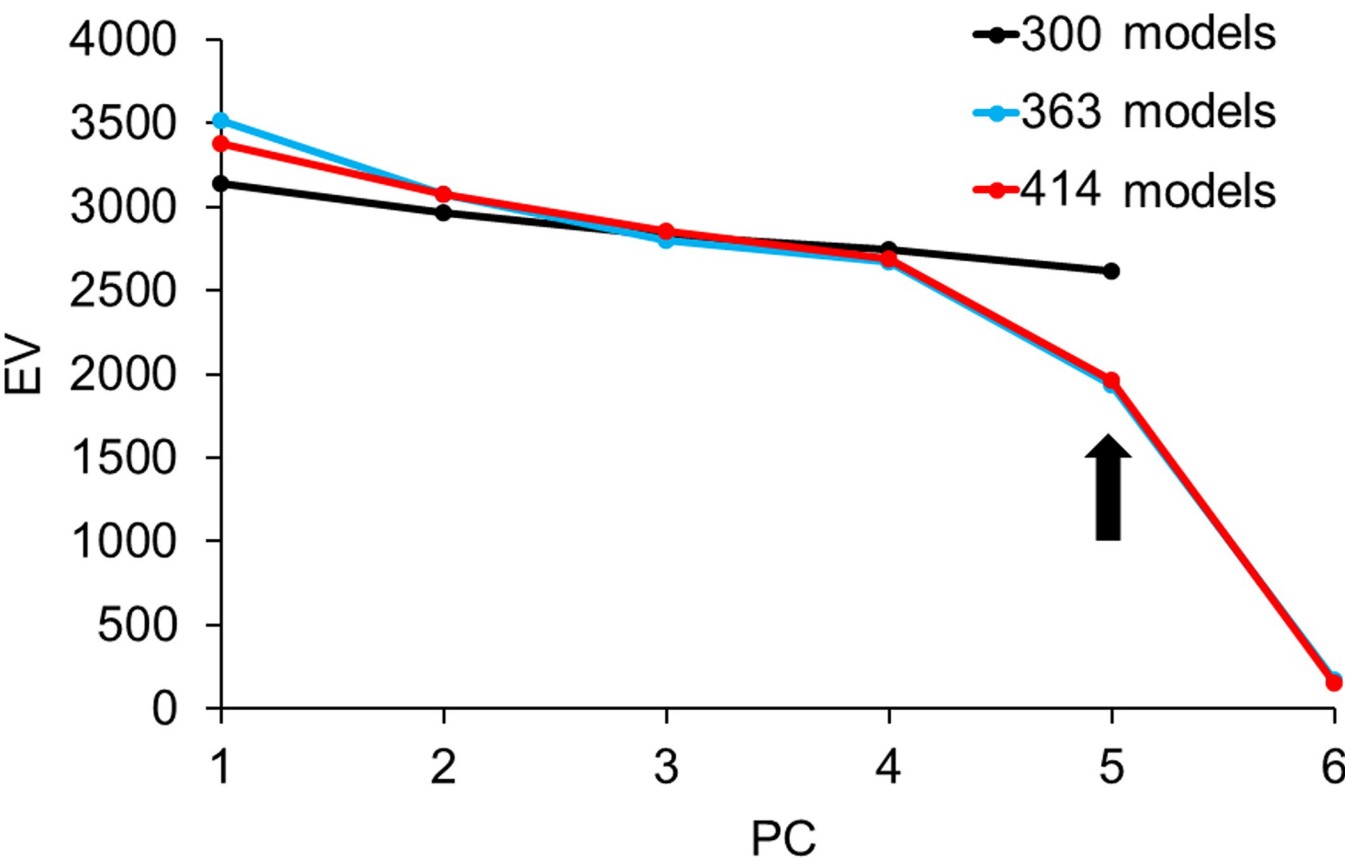

**Fig 13. Scree plot curve of 50 CIs created by each number of models.** PC, principal comportment; EV, Eigen values; CI, composite image; CCI, Classification Criterion Image. EVs strongly decrease at PC6. Since PC6 had a small EV, PC values up to PC5 were used to devise the final CCIs.

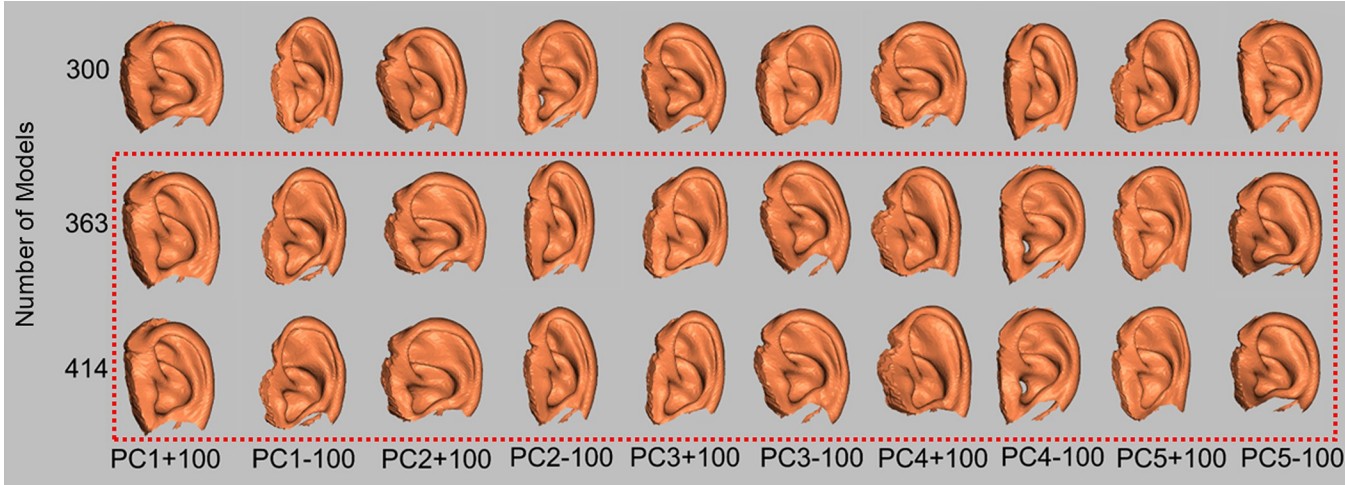

**Fig 14. Ten CCIs for each of the three sets of models of the upper auricle devised from single PC values (±100) from PC1–5 detected by PCA of 50 CIs.** PC, principal comportment; PCA, principal component analysis; CI, composite image; CCI, Classification Criterion Image. Each criterion image of 363 and 414 models is similar to each other. However, CIs from 300 and 363 models showed deviations in their shapes.

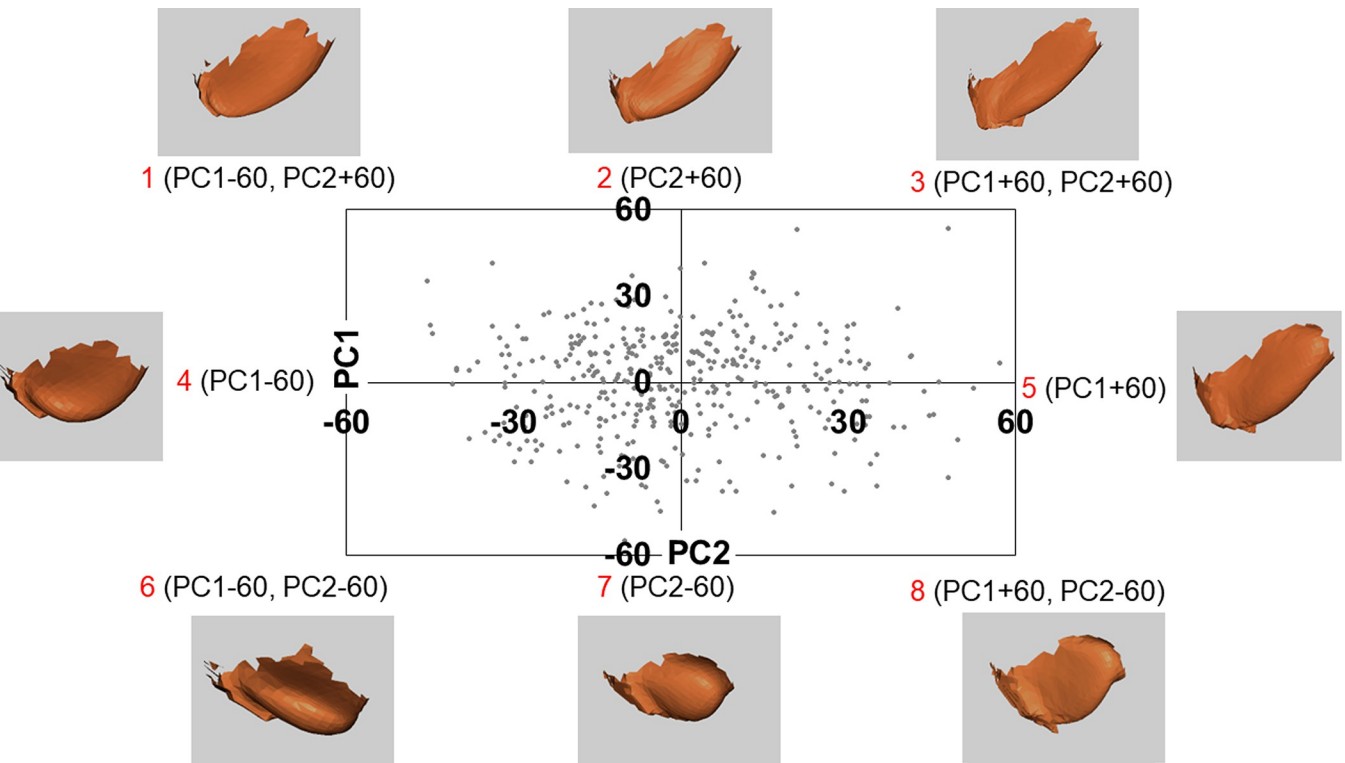

**Fig 15. Biplot of PC1 and PC2 for lobule.** PC, principal component; CI, composite image. Eight lobule models are synthesized using single PC values (−60 and +60) of four (2, 4, 5, 7) and double PC values of four (1, 3, 6, 8) to create the CIs.

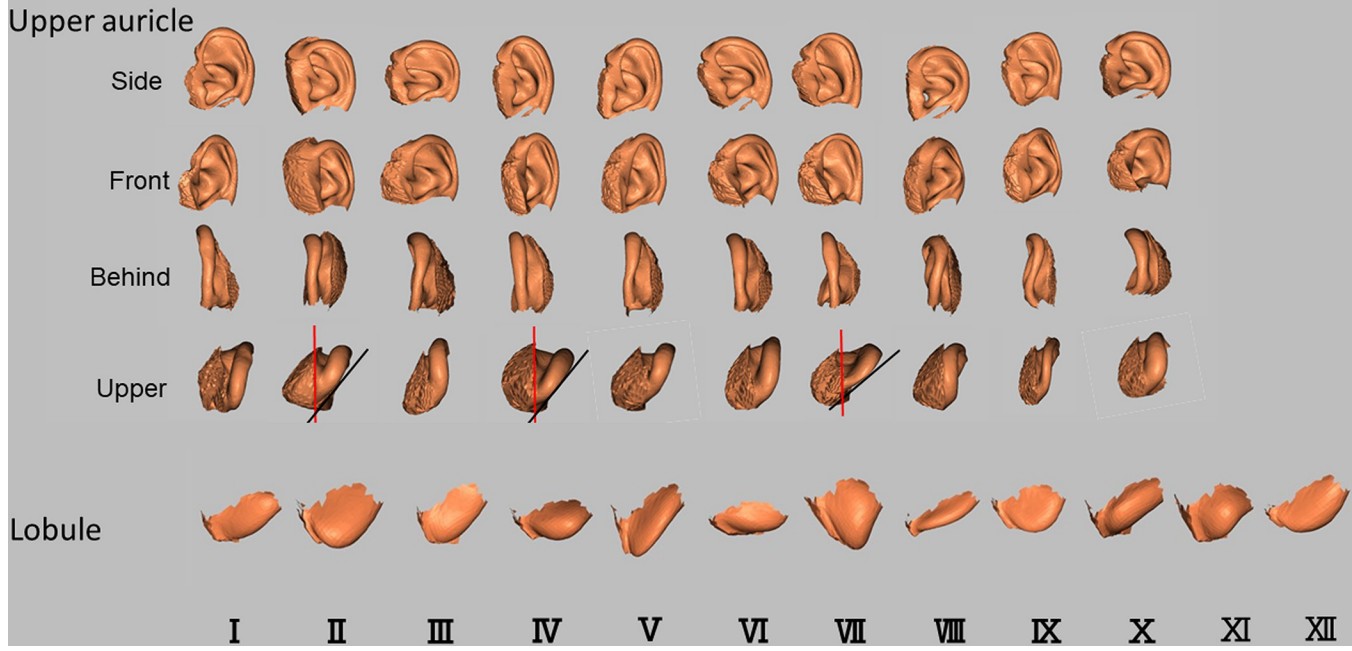

**Fig 16. CCIs made from PCA of 414 ear homologous modeling from the upper auricle and lobule models.** PC, principal component; PCA, principal component analysis; CCI, classification criterion image. Overhanging of the auricle (black line) to the head (red line) is relatively strong in the upper auricles of the upper images of CCI II, IV, and VII.

**Table 5. Details of the results of the CCIs.**

| | Number (%) | |
|---|---|---|
| | Upper auricle | Lobule |
| **Under threshold** | 378 (91.3) | 398 (96.1) |
| **Over threshold** | 36 (8.7) | 16 (3.9) |
| **Multiple classification model detection** * | 263 (63.5) | 310 (74.9) |
| **Asymmetrical** ** | 119/186 (63.9) *** | 142/186 (76.3) *** |

CCI, classification criterion image

*: number of models that detected multiple classified models under the threshold

**: number of models that detected left and right different classified models

***: count from the model that both left and right ear existed.

Previously reported methods for human identification using the ear include the morphological classification of each part of the ear by Moriiyoshi et al. [2] and ear measurement by Ito et al. [21]. The former emphasizes the possibility of determining whether individuals are the same or different by classifying them using characteristics of ear shape from two-dimensional images, whereas the latter analyzes differences due to age and sex by directly measuring ear length, ear width, auricular attachment length, auricular cartilage length, and ear lobule length. The classification of the ear, which has an extremely complex shape, requires multifaceted observation; therefore, evaluation using 3D images to supplement previously reported methods can further improve the accuracy of individual identification. Because forensic analysis using images has recently become a common method, we hope that the CCIs we have

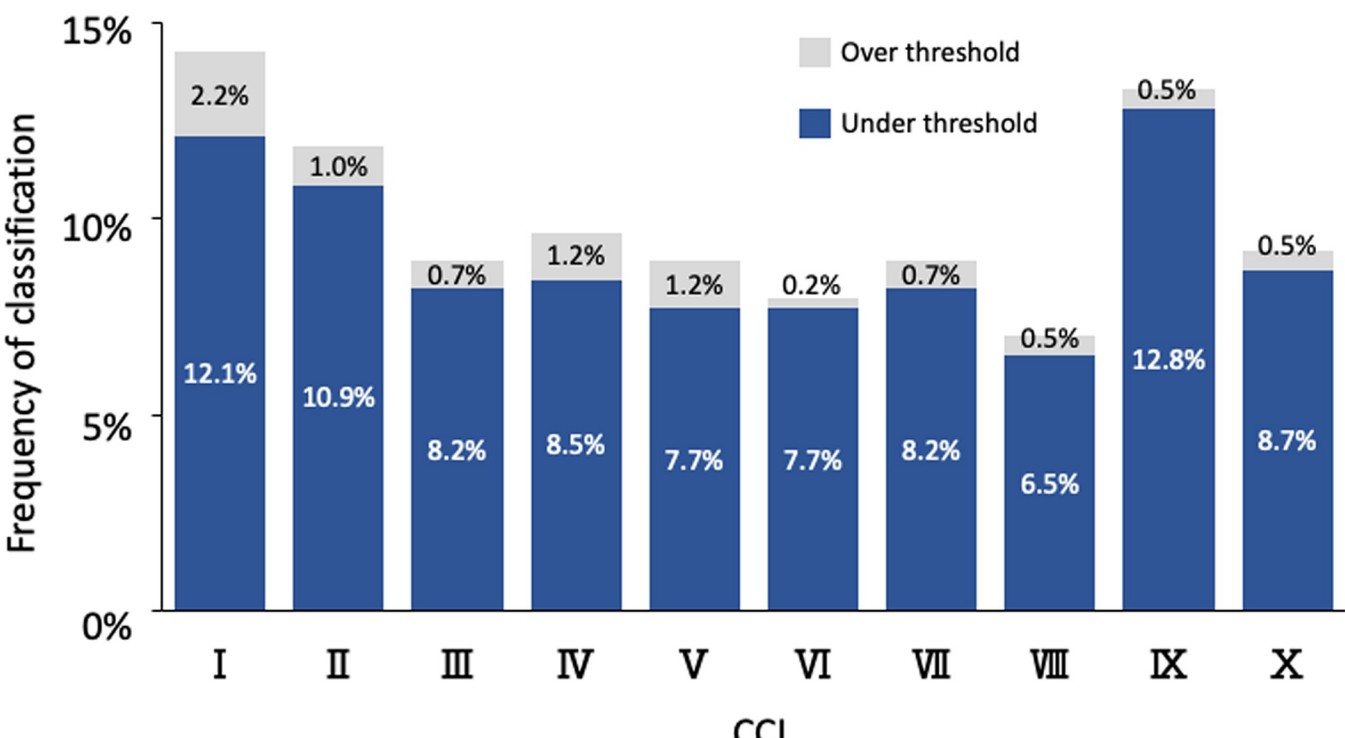

**Fig 17. Results of image determination for 414 upper auricle images by ten CCIs.** CCI, classification criterion image.

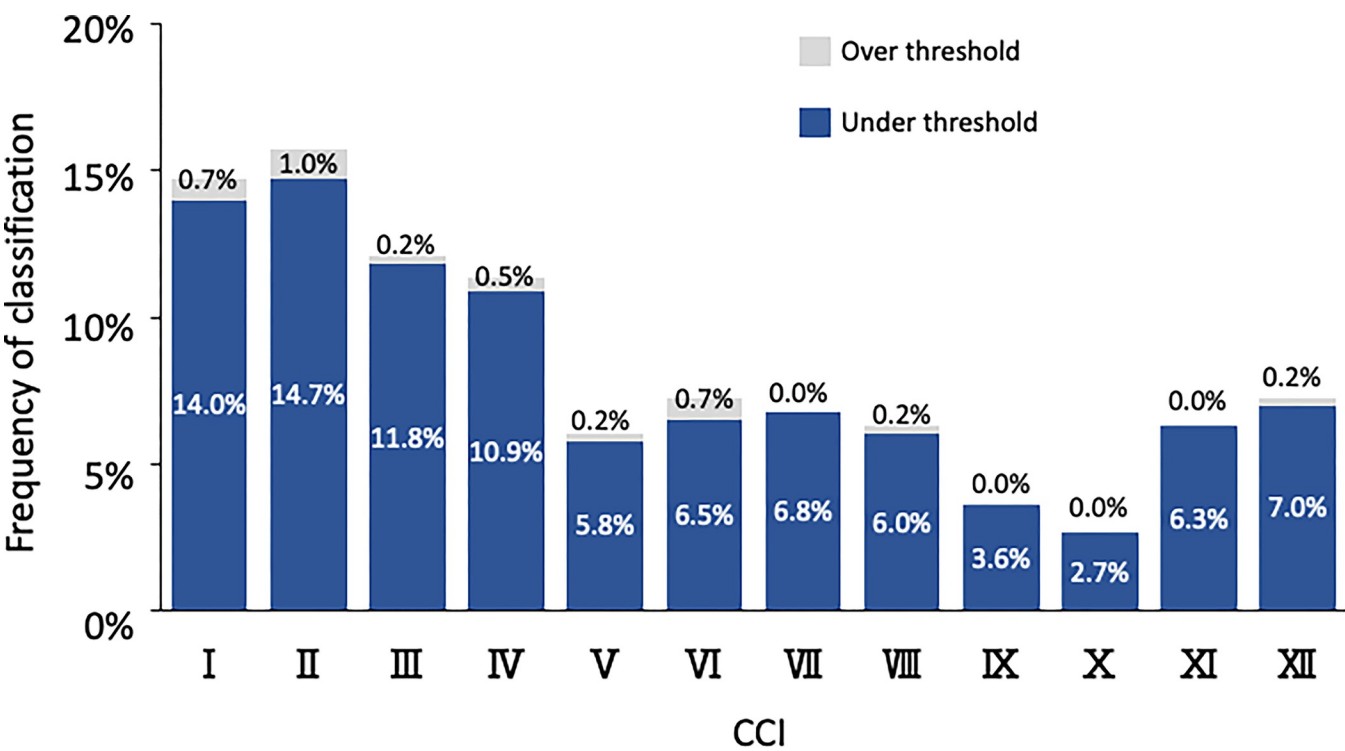

**Fig 18. Results of image determination for 414 lobule images by twelve CCIs.** CCI, classification criterion image.

constructed will be effectively applied and further developed in forensic investigations in the future.

## Supporting information

**S1 Data.**
(ZIP)

## Author Contributions

**Conceptualization:** Hitoshi Biwasaka, Akiko Kumagai.

**Data curation:** Akihito Usui, Masataka Takamiya.

**Formal analysis:** Hitoshi Biwasaka.

**Investigation:** Hitoshi Biwasaka.

**Methodology:** Hitoshi Biwasaka.

**Writing – original draft:** Hitoshi Biwasaka, Akiko Kumagai.

**Writing – review & editing:** Nikolaos Angelakopoulos, Roberto Cameriere, Akiko Kumagai.

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
