## [Decision Letter · Decision Letter 0]

31 Jul 2024

PONE-D-24-23810Validation of morphological ear classification devised by principal component analysis using three-dimensional images for human identificationPLOS ONE

Dear Dr. Kumagai,

Thank you for submitting your manuscript to PLOS ONE. After careful consideration, we feel that it has merit but does not fully meet PLOS ONE’s publication criteria as it currently stands. Therefore, we invite you to submit a revised version of the manuscript that addresses the points raised during the review process.

We look forward to receiving your revised manuscript.

Kind regards,

Mario Milazzo

Academic Editor

PLOS ONE

2. Thank you for stating the following in your Competing Interests section: "The authors declare that they have no known competing financial interests or personal relationships that could appeared to influence the work reported in this paper." 

Reviewers' comments:

Reviewer's Responses to Questions

**Comments to the Author**

1. Is the manuscript technically sound, and do the data support the conclusions?

Reviewer #1: Yes

Reviewer #2: Yes

2. Has the statistical analysis been performed appropriately and rigorously? 

Reviewer #1: Yes

Reviewer #2: Yes

3. Have the authors made all data underlying the findings in their manuscript fully available?

Reviewer #1: Yes

Reviewer #2: No

4. Is the manuscript presented in an intelligible fashion and written in standard English?

Reviewer #1: Yes

Reviewer #2: Yes

5. Review Comments to the Author

Reviewer #1: I would like to commend the authors on a thoroughly done study on ear morphology classification. Not only is the subject of the investigation critical in forensic science and human identification, the authors have alo used sound statistical models to validate their study. I recommend acceptance of the article for the publication as it is.

Reviewer #2: Dear Authors, I sincerely congratulate you both on the research idea and really unique approach for a research question. I think that study presents great contribution to the field of the forensic facial identification, but also to the other forensic fields that have been turning to the analysis of the 3D images, instead of sticking to traditional scorings, measurements, and landmark. The study has been well-planed and conducted, and results and interpretations are provided in scientifically sound manner. However, I think there are certain issues that should be addressed, first to improve the manuscript quality and clarity, and second, to enhance the real impact of the study, particularly regarding its comprehensibility to more general readership (as Plos is highly interdisciplinary journal), its replicability, and application to other similar research question. I will provide my specific suggestions point-by-point below:

1) First, I must reflect to the question defined by reviewer form “Have the authors made all data underlying the findings in their manuscript fully available?”. My response was no, as no data availability statement has been made. I don’t agree with the statement that “All relevant data are within the manuscript” because manuscript reports study findings only. I know these are biological and sensitive data, but since they do not contain complete face that could reveal someone identity and since they are anonymized, I cannot see an obstacle to provide them. I strongly believe that in this particular case is essential to provide something not only because of data itself but because that the software used is closed, i.e., it is not open-sourced and does not allow for other researchers to try to examine their own research question in that way. This being said, I believe although not limitation from research standpoint, it is the greatest factor affecting real impact of the paper, especially in an area of open science policies.

2) I see that the age range is 17-93. Does the ethical approval cover minors?

3) Introduction completely lacks background regarding the main research topic – ears. What is known about that? What are research gaps that are to be filled with present study?

4) M&M – Was there any specifically defined anatomical border to “cut-off” ears?

5) M&M “Each analysis in this study was conducted using varying numbers of participants to determine the optimal number of models required to develop the final CCIs (Table 2)” This sentence is completely clear and correct but wording in the rest of the text “model size” is confusing as it looks like more like size of the model then the sample size. Pleas adjust in whole ms.

6) M&M Verification of accuracies – It is not completely clear what does accuracy mean in this case, please provide more information.

7) Discussion ln. 324-329. This part is appropriate for the introduction only, not discussion. The first paragraph of the discussion should contain main findings and the contribution of the study. How does it contribute to forensic identification?

8) Discussion Study findings should be discussed not only in context of studies that employed same PCA methodology, but also in the context of studies dealing with ears in identification. There is an overemphasis on technical details at the expense of broader interpretations and implication.

9) Discussion – Limitations are too general, they should be more specific. For example, regarding the application in forensic context, can Japanese males be representative for all the populations? What is framework for the future studies and validations when the software is closed?

6. PLOS authors have the option to publish the peer review history of their article (what does this mean?). If published, this will include your full peer review and any attached files.

Reviewer #1: No

Reviewer #2: No

---

## [Author Response · Author response to Decision Letter 0]

23 Sep 2024

Reviewer #1:

 I would like to commend the authors on a thoroughly done study on ear morphology classification. Not only is the subject of the investigation critical in forensic science and human identification, the authors have alo used sound statistical models to validate their study. I recommend acceptance of the article for the publication as it is.

Response:

We are most grateful to the reviewer #1 for your warm-hearted comments on our manuscript.

Reviewer #2: 

1) First, I must reflect to the question defined by reviewer form “Have the authors made all data underlying the findings in their manuscript fully available?”. My response was no, as no data availability statement has been made. I don’t agree with the statement that “All relevant data are within the manuscript” because manuscript reports study findings only. I know these are biological and sensitive data, but since they do not contain complete face that could reveal someone identity and since they are anonymized, I cannot see an obstacle to provide them. I strongly believe that in this particular case is essential to provide something not only because of data itself but because that the software used is closed, i.e., it is not open-sourced and does not allow for other researchers to try to examine their own research question in that way. This being said, I believe although not limitation from research standpoint, it is the greatest factor affecting real impact of the paper, especially in an area of open science policies.

Response.

We understand the reviewer's opinion. However, this article has not concluded that individuals can be determined by the ear shapes. This is a study that devised the classification images of ear for human identification. 

Reviewer #2 pointed out that the classification determination method cannot be verified because other researchers cannot use the same software, and is not practical. However, the software used in this study was applied to minimize error for the frequency of classification, namely, to narrow down the classification. The classification criterion images (CCIs) finally created are possible to classify it sufficiently with macroscopy.

2) I see that the age range is 17-93. Does the ethical approval cover minors?

Response.

The opt-out method ensures that the family members have the opportunity to refuse to participate in the study. Therefore, the ethical approval has been obtained for the study to be applicable to minors as well.

The sentence regarding opt-out method was added to the last of paragraph in the Materials and Methods.

3) Introduction completely lacks background regarding the main research topic – ears. What is known about that? What are research gaps that are to be filled with present study?

Response.

As the reviewer pointed out, the background of the main topic of this research had lacked in the Introduction section. We added the sentences in the Introduction section.

“Even when the face is covered by a mask, the ears can serve as a useful identification feature, as long as they are not hidden by hair.”

“Developing such a classification would allow morphological similarities to be quantified, enabling objective evaluation for human identification.”

4) M&M – Was there any specifically defined anatomical to “cut-off” ears?

Response.

There is not any specific defined. We simply cut ears off with along the lines connecting each anatomical points.

5) M&M “Each analysis in this study was conducted using varying numbers of participants to determine the optimal number of models required to develop the final CCIs (Table 2)” This sentence is completely clear and correct but wording in the rest of the text “model size” is confusing as it looks like more like size of the model then the sample size. Pleas adjust in whole ms.

Response.

As the reviewer pointed out, the explanation “model size” is incur the confusion.

“model size” was changed like to “number of models in set” “set of models” in whole of article.

6) M&M Verification of accuracies – It is not completely clear what does accuracy mean in this case, please provide more information.

Response.

The explanations were insufficient in the section. The subtitle in the Material and Methods was revised to “Verification of the accuracies of CCIs in distinguishing ear morphology”. 

The information was added in the first sentence in this section.

7) Discussion ln. 324-329. This part is appropriate for the introduction only, not discussion. The first paragraph of the discussion should contain main findings and the contribution of the study. How does it contribute to forensic identification?

Response. 

324-239 in the Discussion section have been moved to the Introduction section, and the following sentences were added to the beginning of the discussion.

“Ears, like faces, vary in morphology between individuals … Therefore, we created complex ear CCIs by adopting a new statistical method using 3D images.” 

8) Discussion Study findings should be discussed not only in context of studies that employed same PCA methodology, but also in the context of studies dealing with ears in identification. There is an overemphasis on technical details at the expense of broader interpretations and implication.

Response. 

The sentences were added the last of the Discussion section.

“Previously reported methods for human identification … and further developed in forensic investigations in the future.”

9) Discussion – Limitations are too general, they should be more specific. For example, regarding the application in forensic context, can Japanese males be representative for all the populations? What is framework for the future studies and validations when the software is closed?

Response.

The sentences were added in the second from the last in the Discussion section.

“The subjects of the present study were limited to males … we believe that CCIs that encompasses the auricular features of Japanese males were generated.”

As stated in the answer to Comment 1), the software was used to narrow down the CCIs, and the final judgment was made by the macroscopy, so it is not become the factor affecting real impact of the paper even if the software may close.

---

## [Decision Letter · Decision Letter 1]

4 Oct 2024

Validation of morphological ear classification devised by principal component analysis using three-dimensional images for human identification

PONE-D-24-23810R1

Dear Dr. Kumagai,

We’re pleased to inform you that your manuscript has been judged scientifically suitable for publication and will be formally accepted for publication once it meets all outstanding technical requirements.

Kind regards,

Mario Milazzo

Academic Editor

PLOS ONE

Reviewers' comments:

Reviewer's Responses to Questions

**Comments to the Author**

1. If the authors have adequately addressed your comments raised in a previous round of review and you feel that this manuscript is now acceptable for publication, you may indicate that here to bypass the “Comments to the Author” section, enter your conflict of interest statement in the “Confidential to Editor” section, and submit your "Accept" recommendation.

Reviewer #2: All comments have been addressed

2. Is the manuscript technically sound, and do the data support the conclusions?

Reviewer #2: Yes

3. Has the statistical analysis been performed appropriately and rigorously? 

Reviewer #2: Yes

4. Have the authors made all data underlying the findings in their manuscript fully available?

Reviewer #2: No

5. Is the manuscript presented in an intelligible fashion and written in standard English?

Reviewer #2: Yes

6. Review Comments to the Author

Reviewer #2: I thank authors for accepting suggestions and making required modifications to the manuscript. I think that the manuscript is now acceptable for the publication and that it would present a great contribution to the field. The only thing, that in my opinion has not been well resolved is data availability, but I think the editors will be more competent to decide about it.

7. PLOS authors have the option to publish the peer review history of their article (what does this mean?). If published, this will include your full peer review and any attached files.

Reviewer #2: No

---

## [Editor Report · Acceptance letter]

9 Oct 2024

PONE-D-24-23810R1 

PLOS ONE

Dear Dr. Kumagai, 

I'm pleased to inform you that your manuscript has been deemed suitable for publication in PLOS ONE. Congratulations! Your manuscript is now being handed over to our production team.

Kind regards, 

on behalf of

Dr. Mario Milazzo 

Academic Editor

PLOS ONE